# Zebrafish Models in Neural and Behavioral Toxicology across the Life Stages

**Bruna Patricia Dutra Costa** [1,2], **Layana Aquino Moura** [2], **Sabrina Alana Gomes Pinto** [3],
**Monica Lima-Maximino** [1,4] **and Caio Maximino** [1,2,*]

[1] Biodiversity and Biotechnology Network of the Legal Amazon, Adrianópolis Manaus 69057-070,
AM. CEP, Brazil; costabruna28@gmail.com (B.P.D.C.); monica.lima@uepa.br (M.L.-M.)

[2] Laboratory of Neurosciences and Behavior, Institute for Health and Biological Studies, Federal University of
the South and Southeast of Pará, Marabá 68507-670, PA, s/ CEP, Brazil; layanamoura12@gmail.com

[3] Faculty of Psychology, Institute for Health and Biological Studies, Federal University of the South and
Southeast of Pará, Marabá 68507-670, PA, s/ CEP, Brazil; sabrina-alana@outlook.com

[4] Laboratory of Neuropharmacology and Biophysics, Center for Biological and Health Sciences, Pará State
University, Marabá 68507-670, PA, CEP, Brazil

[*] Correspondence: cmaximino@unifesspa.edu.br

**Abstract:** The industry is increasingly relying on fish for toxicity assessment. However, current guidelines for toxicity assessment focus on teratogenicity and mortality. From an ecotoxicological point of view, however, these endpoints may not reflect the "full picture" of possible deleterious effects that can nonetheless result in decreased fitness and/or inability to adapt to a changing environment, affecting whole populations. Therefore, assessing sublethal effects add relevant data covering different aspects of toxicity at different levels of analysis. The impacts of toxicants on neurobehavioral function have the potential to affect many different life-history traits, and are easier to assess in the laboratory than in the wild. We propose that carefully-controlled laboratory experiments on different behavioral domains—including anxiety, aggression, and exploration—can increase our understanding of the ecotoxicological impacts of contaminants, since these domains are related to traits such as defense, sociality, and reproduction, directly impacting life-history traits. The effects of selected contaminants on these tests are reviewed, focusing on larval and adult zebrafish, showing that these behavioral domains are highly sensitive to small concentrations of these substances. These strategies suggest a way forward on ecotoxicological research using fish.

**Keywords:** neurobehavioral assessment; ecotoxicology; zebrafish; neurotoxicology

---

## 1. Introduction

Ecotoxicology, as a field, currently experiences an influx of research using fish as subjects, in part due to increased interest in the toxicology of pesticides and waste from pharmaceutical products, which end up in water bodies and affect aquatic life. While ecotoxicology usually focuses on toxic effects at the population, community, ecosystem, and biosphere levels, the integrative approach usually demanded to reach this level of analysis implicates assessment of effects of the toxicants across all levels of biological organization [1]. As a result, all effects that are likely to result in decreased fitness and/or inability to adapt to a changing environment are of relevance. Clearly, the impacts of toxicants on neurobehavioral function have the potential to influence many different domains of life [2,3]. Consider, for example, effects on defensive behavior: contaminants that change the appropriate levels of antipredator defense and cautious exploratory behavior can lead to a decreased ability to escape or avoid actual or potential threats [4], causing either death or the loss of important resources,

such as access to mates or food. Examples of contaminants that have been shown to change cautious exploratory behavior include atrazine [5], methylmercury [6,7], PCB126 [8], dimethyl sulfoxide [9], and copper [10]. Ammonia [11] and IPBC (3-iodo-2-propynyl-N-butyl carbamate) [12] have been shown to impact antipredatory and alarm responses.

The field of ecotoxicological research using fish species gained much traction in the last 20 years [13]. While behavioral testing has been common for a longer period (at least since the 1970; e.g., Rand's 1985 review [14]), the increased availability of laboratory assays and protocols, as well as pressures from special interests groups and increasing awareness from regulatory agencies that lethal endpoints are not sufficient for ecotoxicology, have brought renewed interest in the field in a way that sublethal behavioral and physiological effects are now more common in the area than the usual protocols [15]. Organisation for Economic Co-operation and Development (OECD) guidelines for assessing the toxicity of chemicals in fish include protocols with lethal endpoints (for acute toxicity in adults [OECD 203], embryo toxicity [OECD 210 and 236], and toxicity tests on egg and sac-fry stages [OECD 210 and 212]), as well as sublethal endpoints (larval and juvenile growth [OECD 210 and 215] and sexual and endocrine development [OECD 230 and 234]). However, the field as a whole is rapidly moving beyond mortality and teratogenicity. The present paper reviews selected references in the field of neurobehavioral ecotoxicology research, proposing the use of a handful of sensitive and ecologically relevant behavioral assays to expand beyond lethal and/or crude morphological endpoints. The paper begins by dispelling confusions on the use of fish as both model and target species in ecotoxicological research. The paper then moves on to the core of the article, namely, biobehavioral assays in neurotoxicology.

## 2. Using Fish as Models and Targets in Ecotoxicology

Fish are directly affected by a plethora of environmental toxicants, including heavy metals from mining, pesticides, and pharmaceutical waste from human consumption. The effects of a contaminant can be studied using fish behavior as a toxicological indicator, with the objective of understanding impacts on natural populations. In that case, wild species are best suited, preferably derived from wild populations from areas that are contaminated.

In many cases, however, fish are used as model organisms—that is, as a surrogate species "that are extensively studied in order to understand a range of biological phenomena, with the hope that data, models and theories generated will be applicable to other organisms, particularly those that are in some way more complex than the original" [15], p. 209. Model organisms are used to study a wide range of systems and processes occurring in living organisms, including genetics, development, physiology, behavior, evolution, and ecology [16,17]. Usually, model organisms are selected based on experimental characteristics that facilitate its use in molecular and physiological research [17]: small sizes; low costs to breed, maintain, and transport; short generation times and life cycles; high fertility rates; high susceptibility to techniques for genetic modification. In addition to that, infrastructure (including shared databases and tools) and a well-established community of researchers that rely on the species as a model organism are also fundamental [17].

Fish can be used as model organisms in ecotoxicology in at least two important contexts: the first is of using a fish as a model organism to infer processes and effects on other fish species. For example, organic mercury intoxication in the Amazon, due to bioaccumulation, highly impacts piscivorous fish [18] that are hard to raise in laboratory contexts, and to which neurogenomic and behavioral tools are unavailable. Using smaller fish species—including well-established model organisms such as zebrafish and medaka [19]—can circumvent these difficulties if one assumes that the physiology of teleost fish is relatively well-conserved across major fish taxa [20].

While using fish as model organisms in ecotoxicology is important to assess the impacts of contaminants on aquatic organisms, they can also be used to try to predict toxic effects on non-fish species, including humans. In many well-established protocols for teratogenicity and reproductive toxicology, the assumption appears to be that, if a given substance is teratogenic in fish embryos, one can assume that it will also be in human embryos (e.g., [21,22]). While the physiological distance between

fish and humans is certainly higher than among teleosts, or even to non-teleosts, the assumption is that there are enough commonalities to facilitate extrapolation [13,19,23]. Moreover, from a mechanistic and evolutionary point of view, the discovery of a mechanism of toxicity that is shared between non-human mammals and fish make it more likely that the mechanism is evolutionarily conserved, and therefore also shared with humans [24].

## 3. Behavioral Bioassays in Neurotoxicology

Acute toxicity assays, such as OECD protocols (acute toxicity in adults [25] and the fish embryo test [26]) and fish acute toxicity syndromes [27–29], are important from the point of view of policies and regulations, as most regulatory agencies use them as legal bases to determine whether and how the chemicals can be put into the market. These assays, whether using lethal or sublethal endpoints, usually focus on dramatic effects that lead to a very high lethality. However, there are many other toxic effects that are relevant for ecotoxicological research, including its consequences on protocols for screening chemicals [2,30]. Along with other measures of sublethal toxicity that have a long history in fish toxicology (e.g., physiological parameters in blood and other body fluids [31], histological parameters [32], and biochemical parameters [33]), behavior can be used as an assay of fitness [34], given its importance to the viability of the organism, the population, and the community. Peterson et al. [30] pointed that toxicants can produce effects at individual-level responses, which in its turn impact responses at the population, species, community, ecosystem, and evolutionary levels (Figure 1).

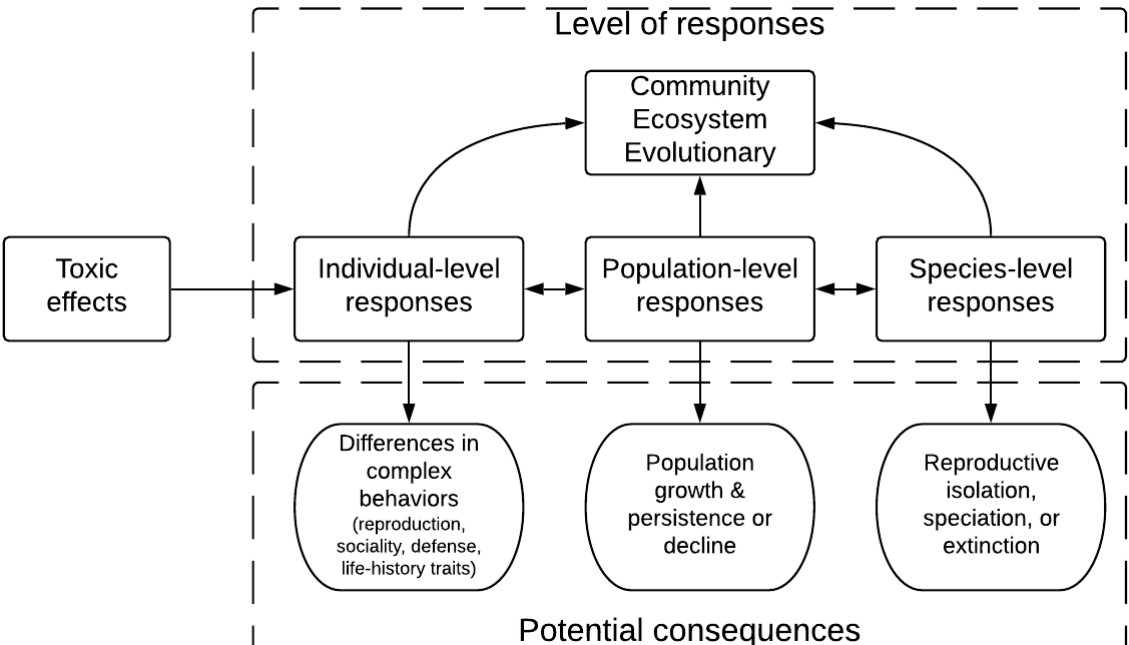

**Figure 1.** Toxic effects can act at different response levels (individual, population, species, community, ecosystem, or evolutionary levels) to produce different consequences on fitness. Behavioral effects impact not only the individual level, but its interaction with other levels as well.

There are several advantages of incorporating behavior in ecotoxicology studies [30]:

1.  Behavior is an indicator of multiple levels of biological effects [14,30].
2.  Behavior is among the most sensitive indicators of impact of exposure, with some estimates putting it as 10–1000 times more sensitive than lethality measures [30,35].
3.  Behavior is considered an early warning tool [2].

One difficulty of incorporating behavior, however, is that of ecological validity. In psychometry, ecological validity is a measure of how test performance predicts behaviors in real-world settings; in the case at hand, the highest possible ecological validity is observing behavior in the wild. This is, in many cases, impractical, and does not lend itself to mechanistic ecotoxicology, as analyzing the effects of single toxicants, or knowing precise concentrations, is impossible, and avoiding contamination is very difficult.

One solution is to develop behavioral bioassays. These "behavioral bioassays measure an organism's behavior, qualitatively or quantitatively, to detect and analyze some external stimulus or as an indicator of an internal physiological or psychological state" [36]. In the literature, most assays are described as related to behaviors that are associated with mental disorders—anxiety-like behavior, aggressive behavior, and compulsive-like behavior; however, these assays allow broader comprehension of a mechanism, without a necessary causal analogy with the etiology or pathological basis of the mental disorder [37]. As a result, behavioral bioassays are used to study normal behavior and/or the effects of perturbations on this trait [38].

In fact, most of the so-called "animal models" in fish research [39] are not models per se, but screening tests or behavioral bioassays [37]. These models usually present good predictive validity, as they discriminate between drugs with clinical efficacy (e.g., ref. [40]). The predictive validity of most toxicity assays to discriminate between toxic effects and/or mechanism of action, however, is not yet established. This represents both an opportunity and a difficulty for research: part of the necessary steps in introducing behavioral bioassays in aquatic ecotoxicology involves establishing predictive validity and conducting mechanistic research for different assays—i.e., establishing how specific toxicants induce predictable neurobehavioral and physiological responses at different levels.

A corollary of predictive validity, of course, is that the model should be able to predict real world exposures and consequences. Most laboratory tests do not use real-world conditions; laboratory experiments isolate toxicants instead of using mixtures, often fail to use concentrations that are relevant in the wild, and focus on behaviors that are easy to manipulate or elicit. This is further discussed in Section 6.

In addition to using behavioral assays that directly mimic challenges faced in the wild, such as conspecific conflict, foraging, and antipredator defense, it can be useful to assess behavior in assays that carefully target the neurobehavioral domains that are recruited during these challenges. This is the "ethoexperimental" approach [41] that we will follow in this review. For zebrafish and other small teleost species, there are currently well-validated assays to study sociality [42], anxiety-like behavior [43], aggression [44], and fear [45], as well as appetitive learning [46], which is relevant to foraging (Figure 2). While ecological validity is diminished by analyzing behavior at a more "basic" level of a complex, composite function, this is offset by greater experimental control and ability to manipulate and test stimulus control.

In the remainder of this review, we will focus on these behavioral assays and how these have been used in studying the effects of toxicants. Specific focus will be given to locomotor activity, which is relevant to many different domains, including defensive behavior, foraging, and habitat use; fear and anxiety, which is relevant for antipredator defense; and aggression, which is relevant to defense against conspecifics and territoriality.

*3.1. Locomotor Activity Assays*

3.1.1. Larvae

Locomotor activity is sensitive to many different perturbations, and is organized at different levels of the nervous system [47,48]. Alterations in locomotor activity, therefore, can represent modifications at the neuromuscular junction, in muscle, at central pattern generators in the spinal cord or in the hindbrain, or in more rostral levels [49]. While highly sensitive to perturbations, usually assays for locomotor activity focus on distance or speed [50], which are not very selective effects.

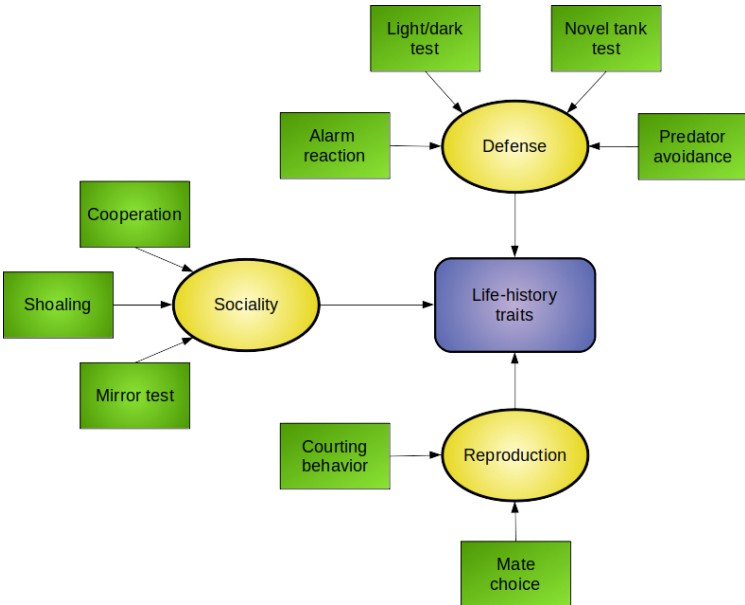

**Figure 2.** Well-controlled laboratory experiments, using more than one test, can determine whether the effect of a given toxicant impacts one or more behavioral domains that are likely to affect complex responses (reproduction, defense, sociality) that are related to life-history traits, and likely to impact responses at population, species, community, and ecosystem levels.

Fish larvae show greater activity during dark cycles than during light cycles; thus, larvae respond to changes in illumination with both acute responses and extended behavioral responses [51,52]. The effects of a wide range of compounds on the locomotion of zebrafish larvae have been tested, and its acute effects have been classified crudely as sedative-like or stimulant-like (Table 1). As can be seen in Table 1, if one analyzes only simple endpoints, such as the distance travelled or swim speed, the predictive validity is very low—for example, cocaine and amphetamine clearly do not produce sedative-like effects in other animals.

**Table 1.** Examples of agents with sedative-like and stimulant-like effects on larval locomotor behavior. Sedative-like effects represent decreased swimming speed and/or distance, in either a light or dark period, while stimulant-like effects represent increased swimming speed and/or distance. Concentrations have been standardized to molarity to facilitate comparisons. hpf = hours post-fertilization; dpf = days post-fertilization.

| Class | Compound | Concentration Range | Age | Ref. |
|---|---|---|---|---|
| Sedative-like | 4-aminopyridine | 0.6 mM | 5 dpf | [53] |
| | Clozapine | 12.5–50 mM | 7 dpf | [54] |
| | Cocaine | 0.2–50 μM | 6 dpf | [47] |
| | Amphetamine | 0.1–20 μM | 6 dpf | [47] |
| | Diazepam | 10–100 nM | 7–14 dpf | [47] |
| | Fluoxetine | 4.6 mM | 3–6 dpf | [55] |
| | Polybrominated diphenyl ether (PBDE-71) | 63.8136 nM | 5 dpf | [56] |
| | Graphene quantum dots (light period) | 1.0408–16.6528 mM | 4–96 hpf | [57] |
| | Silica nanoparticles (dark period) | 416.1119 and 832.2237 nM | 4–96 hpf | [58] |
| | Perfluorooctanesulphonic acid (PFOS) | 0–15.9958 μM | 6–120 hpf | [59] |
| Stimulant-like | Silica nanoparticles (dark period) | 1.664 and 3.328 μM | 4–96 hpf | [58] |
| | 4-Aminopyiridine | 0.8–2.5 mM | 5 | [53] |
| | Aconitine | 2.5–25 μM | 5 | [53] |
| | Bisphenol A | 0.01–1 μM | 5 | [60] |
| | Pentylenetetrazole | 10 mM | 5 | [53] |

As a result of this lack of specificity in locomotor activity assays in fish larvae, many different tests have been developed, and a vast majority of these used zebrafish as a model organism [52]. Zebrafish larvae exhibit different behaviors to different stimuli, which can be exploited to assess different neurobehavioral domains (Table 2). These responses allow a much more detailed investigation of locomotor behavior at earlier life stages and with ecologically-relevant endpoints.

**Table 2.** Additional behavioral endpoints/assays in fish larvae that can be exploited for increased sensitivity and/or specificity in ecotoxicological research.

| Behavior | Age at Observations | Stimulus | Example Compound |
|---|---|---|---|
| Coiling | 17–21 hpf | None | Perfluorooctanesulfonic acid (PFOS ) [59] Chlorpyrifos [48] |
| Touch-induced escape responses (touch response) | 22–27 hpf | Touch | Dichlorodiphenyltrichloroethane (DDT) [61] Dieldrin [61] Fipronil [61] Nonylphenol [61] |
| Optokinetic response (OKR) | 73–80 hpf | Moving objects | Digoxin [62] Gentamicin [62] Ibuprofen [62] Minoxidil [62] Quinine [62] |
| Optomotor response | 5 dpf | Moving objects | Bisoprolol [63] Chlorpromazine [63] Cisapride [63] Cisplatin [63] Gentamicin [63] Nicotinic acid [63] Quinine [63] |
| Startle responses | 5 dpf onwards | Vibrational or acoustic stimuli | Lead [64] Mercury [65] |
| Shadow response | 8 dpf | Looming shadows | 2,2′,4,4′-tetrabromodiphenyl ether (BDE-47) [66] |
| Prey capture | 9 dpf | Prey | 2,4-Dichlorophenoxyacetic acid (2,4-D) [67] |

### 3.1.2. Adults

Most assays for locomotor behavior in larvae focus on changes in speed or distance [50]. However, neurotoxic effects can frequently be observed as changes in posture or form of swimming movements in both larvae and adults [68,69]. More fine-grained assays for larvae were described above (see Table 2). However, adult behavior is putatively more complex than larval behavior, and changes in movement patterns are more readily observable due to their larger size. Little and Finger demonstrated that the lowest toxicant concentration that produced changes in adult locomotor behavior lies between 0.1% and 5.0% of the lethal concentration [68]. However, speed and distance are not the only possible endpoints, nor the most sensitive; behavioral endpoints that can be quantified through movement analysis include acceleration, turning angles or frequency, time spent in different swimming modalities (normal swimming, large movement swimming, small movement swimming, burst swimming, etc.), horizontal and vertical distribution of individuals, path tortuosity, and startle responses [69].

### 3.2. Anxiety-Like Behavior

Many options exist currently to assess anxiety-like behavior in fish, most of them using zebrafish [43,70]. The novel tank and light/dark preference tests involve measures of spatio-temporal distribution (time at the bottom of a novel tank or at the dark portion of a light/dark tank) and ethogram (freezing, erratic swimming, risk assessment, thigmotaxis) that are sensitive to anxiolytic or anxiogenic treatments [40]. In a recent meta-analysis, we have shown that the light/dark test is more sensitive to treatments in general than the novel tank test, and that both tests elicit a significant cortisol response

equally [40]. It has been observed that for standardized behavioral practices within the laboratory, such experimental conditions (test days and batches of fish) may have relatively few experimental effects on the outcomes of anxiety and locomotor activity [71]. Examples of toxicants that have been shown to affect anxiety-like behavior in the zebrafish light/dark and novel tank tests can be found on Table 3.

**Table 3.** Examples of toxicants that affect anxiety-like behavior in zebrafish behavioral bioassays. Concentrations have been standardized to molarity to facilitate comparisons.

| Toxicant | Concentration/Dose Range | Duration of Treatment | Test | Ref. |
|---|---|---|---|---|
| Atrazine | 23.1825 nM–14.4891 µM | 4 weeks | Light/dark test (LDT) | [5] |
| Methylmercury | 1–5 mg/kg | Acute | LDT, Novel tank test (NTT) | [6,7] |
| PCB126 | 0.3–1.2 nM | Developmental (4–24 hpf) | NTT | [8] |
| Dimethyl sulfoxide | 7.05 mM | Acute | NTT | [9] |
| Copper sulfate | 60.0769 nM | Acute | LDT, NTT | [10] |

Alterations in anxiety-like behavior are relevant not only to antipredator defenses, but also to foraging and resource finding: if an animal is "too cautious" (i.e., increased anxiety-like behavior) due to the effects of a toxicant, it can miss important opportunities to reproduce or to forage outside its home range. Conversely, decreased anxiety-like behavior can lead to "reckless" behavior that ends in being attacked by a predator, as in migrating salmon smolts [4]. The standardization offered by these behavioral assays can help researchers identify alterations in these endpoints, which can lead to novel hypotheses on the ecotoxicological sublethal effects of substances at complex behaviors.

*3.3. Aggression*

Agonistic and aggressive behaviors are associated with territory defense, in agonistic interactions within a social group, in contests for mate access or food, as well as in prey capture and antipredator behavior [72]. In agonistic interactions, fights are usually substituted for ritualized activity (aggressive displays) through which one of the contestants show its superiority without the need to hurt or kill its opponent, or to hurt itself; agonistic interactions can be an appetitive element of aggression in that it can escalate to actual aggressive behavior, or towards a resolution [73].

Aggressive-like behavior has been studied in laboratory fish with different approaches, ranging from mirror tests (in which an aggressive display are elicited by mirror images) to dyadic fights and group social interaction [44]. The advantage of using the mirror test is that it can capture most elements of aggressive motivation without unnecessarily risking damage to the animals; however, since no resolution is possible in mirror-elicited displays, the full range of behaviors and physiological adjustments is not captured. Choosing between these alternatives involves balancing ecological validity, throughput, and welfare concerns [44].

Alterations in aggressive behavior can potentially decrease fitness by increasing the likelihood of losing a contest, getting damaged after inadequately escalating the fight, or losing access to resources such as territories or food. Table 4 represents some examples of the effects of toxicants in aggressive and agonistic behavior in zebrafish.

**Table 4.** Examples of toxicants that affect aggressive and agonistic behavior in zebrafish behavioral bioassays. Concentrations were converted to molarity to facilitate comparisons.

| Toxicant | Concentration/Dose Range | Duration of Treatment | Test | Ref. |
|---|---|---|---|---|
| 17α-ethinylestradiol | 1.6869–168.6893 pM | 48 h | Dyadic interaction | [74] |
| | 13.4951–74.2233 pM | 14 days | Group interaction | [75] |
| Tetrabromobisphenol A (TBBPA) | 5–50 nM | Developmental (1–120 dpf) | Mirror test | [76] |
| Methylmercury | 4.6376–69.5636 nM | 32 h | Mirror test | [77] |
| Paraquat | 20 mg/kg | 6 injections for 16 days | Mirror test | [78] |

## 4. Moving from Behavioral Toxicology to Ecotoxicology: Ecologically-Relevant Endpoints

The tests and behavioral bioassays that were reviewed in Section 2 are all sensitive to subtoxic concentrations of important environmental contaminants. The tests also have the advantage of being easy to implement in carefully controlled laboratory environments; differently from behavior observed in the field, variables in the laboratory can be cautiously manipulated to produce the most reliable and sensitive measurements that can satisfy regulatory agencies. However, it is not always clear how specific endpoints (e.g., caudal fin tremors [79]) are related to responses at the individual and higher levels, which are of interest to ecotoxicology.

Almost 35 years ago, Rand [14] suggested that the behavioral responses that are more useful for toxicology include those that are (A) well-defined and practical to use; (B) sensitive to a range of contaminants and observable in different species; (C) with known environmental factors; and (D) ecologically relevant. Among these criteria, the first three are amenable to laboratory testing, while the last is usually hypothetical. Increasing ecological relevance can be reached by at least two approaches: the use of neurobehavioral domains, which imply the need to run more behavioral tests; and analyzing the relationship between behavior in the laboratory and behavior in the wild.

The concept of a behavioral domain is widely recognized in behavioral genetics of knockout and mutant laboratory animals [80], in which recognizing whether the effect of a given genetic manipulation is specific to the test or generalizes to a more general domain is important. Behavioral domains of interest to neuroscientists (and, as an extension, to neurotoxicologists) include anxiety, mood, social behavior, cognition, and impulse control. Many of these domains are, hypothetically, related to the complex behaviors observed at the individual level that are shown in Figure 1. Thus, well-controlled laboratory experiments, using more than one test, can determine whether the effect of a given toxicant impacts one or more behavioral domains which are likely to affect these complex responses (Figure 2).

This approach can generate powerful hypotheses that can be further tested in field experiments by toxicologists and ethologists alike, either by direct observation in the wild or by "laboratory in the field" approaches. For example, following observation of natural antipredator behavior in the wild, animals can be captured and taken to the laboratory, and their behavior in standardized assays can be tested to check whether the assay predicts performance in the wild. Although powerful, this approach is unlikely to be of direct interest of neurotoxicologists.

## 5. Sensitivity and Specificity of Behavioral Tests

If relying on carefully controlled behavioral bioassays in the laboratory can increase the sensitivity to find toxic effects [30], two issues arise: the first is the problem of specificity—that is, are the behavioral effects that are observed due to neurotoxicity, or due to nonspecific effects on other systems (e.g., changes in respiration rates or acid-base equilibria)? The second is the problem of using data obtained from laboratory studies to extrapolate to field contamination and, as a result, to defining threshold concentrations that are "acceptable" [81]. The first is a problem of internal validity; the second of external validity. While not the focus of the current review, these problems are important and must be addressed if the full power of using zebrafish as a model organism in neurotoxicology and behavioral toxicology is to be harnessed.

The problem of specificity is less of a problem to ecotoxicology than it is for (mechanistic) neurotoxicology. Toxicants can alter the function of many other systems and not directly affect the nervous system, and nonetheless affect behavior [82]. For example, changes in gill physiology can lead to hypoxia, which in its turn lead to surfacing behavior [83]. The possible consequence of this behavior for the individual is increased probability of attack by aerial predators; thus, whether the toxic effect was produced at the gills or at the brain is inconsequential to this. However, surfacing behavior can be interpreted, in the context of the novel tank test, as decreased anxiety-like behavior [40]. While this distinction appears inconsequential if one looks at the final consequence, it is important for mechanistic research, and can have consequences for proposing mitigation strategies, for example.

The issue of specificity is related to increased sensitivity. Taking a lead from the statistics of medical screening, a test (or battery of tests) that is highly sensitive but with low specificity is likely to produce many false positives, while tests that show low sensitivity but high specificity are likely to produce many false negatives [84]. Thus, if a behavioral test is much more sensitive than a test of mortality and/or teratogenicity to detect a toxic effect, it is also at least theoretically possible that these tests also increase the probability of finding nonspecific effects ("false positives" in terms of neurotoxicology). The most viable solution to this issue, of course, is that widely used in behavioral genetics since the introduction of gene knockouts: to use a battery of tests measuring similar traits, discarding nonspecific effects and thus increasing convergent validity [37].

One of the most common nonspecific toxic effects that is observed in aquatic toxicology is that of nonspecific stress responses. We are defining nonspecific stress responses as the effect of toxicants on behavior and physiology that may resemble stress responses (e.g., increased bottom-dwelling or surfacing, decreases in swimming activity, increases in cortisol), but are in fact indirect responses due to toxicity at non-nervous organ systems. One important example is the effect on gill physiology—not only on gas exchange mechanisms, but also disturbing ion transport processes across the gills. Typical behavioral effects include decreased swimming activity and erratic swimming (e.g., [85–87]), and altered cortisol responses may appear in the absence of toxicant uptake in the hypothalamus-pituitary-interrenal axis (e.g., [88]). This is far from representing a "false positive," but the neuroendocrine and behavioral effects are secondary to the effects on gill physiology. Nonetheless, the secondary behavioral alterations can be as important as the changes on gill physiology in terms of fitness—but care must be taken before assuming neural effects.

Another nonspecific effect that can be observed is related to external validity—that is, whether or not a potentially neurotoxic effect that is observed on behavior is in fact biologically or ecologically meaningful [81,82]. This issue is important because ecotoxicology is not insulated from the regulatory branch, and results from the area are expected to inform policies and regulations on environmental contamination [81,89]. From the point of view of regulators and risk assessors, increasing sensitivity can also increase the rate of false positives, which pose two economical costs: (1) the community mobilizing resources to an environmental problem that does not in fact exist, therefore diverting these resources from more urgent needs; and (2) reiterated false positives ("crying wolf") leading to attitude changes in the community, which ignore future environmental problems [81]. Whether or not these behavioral tests translate into better understanding (fewer "false positives" in terms of ecotoxicology) is an issue that carries both an ethical and an empirical dimension. The ethical dimension is beyond the scope of this manuscript (but see refs. [90–92] for a discussion). The empirical dimension involves focusing not (only) on sensitivity, but also on the predictive capabilities of toxicity screens [93]. This, of course, involves extensive validation, which appears, at the time, to be missing from neurotoxicology research using zebrafish behavior.

## 6. Conclusions

In summary, the power of behavioral experiments in the laboratory can be tapped to expand the reach of ecotoxicological approaches by increasing the number of tests with a domain-based mindset. This does not mean that lethal (acute) tests should be abandoned, given their importance in regulatory agencies and policies regarding, e.g., pesticides. However, using behavioral tests (including the zebrafish bioassays briefly described in this review) can add to the options of sublethal tests that are not only more sensitive than lethal tests, but more likely to detect biologically relevant effects that can affect the mesocosm (population, species, community, ecosystem). The long history of research on sublethal toxicity in fish, and the uneasy relationship between the fields that compose behavioral ecotoxicology (ethology, behavioral neuroscience/pharmacology, and toxicology), need to be put in an integrative framework. To do that, the problem of specificity and the problem of translatability (that is, whether increased sensitivity and specificity translate to better predictions of effects at the population and higher levels, and whether these predictions impact environmental policies) need to be addressed

both at the ethical and empirical frames. The hope of using the full potential of the zebrafish can only be approached if these steps are taken first.

**Author Contributions:** B.P.D.C., L.A.M., S.A.G.P., M.L.-M., and C.M. contributed equally to the first draft of the manuscript. M.L.-M. and C.M. contributed to the further revisions of the manuscript. All authors have read and agreed to the published version of the manuscript.

**Funding:** This research received no external funding.

**Conflicts of Interest:** The authors declare no conflict of interest.

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
