# Peer review of "Zebrafish Models in Neural and Behavioral Toxicology across the Life Stages"

_fishes, doi:10.3390/fishes5030023_

Round 1
Reviewer 1 Report
Overall this is an interesting review of toxicological methods, and the authors provide a convincing argument for using alternative procedures than those currently popular. Most of my comments are relatively minor in nature and deal with structure of the review, rather than its contents which I feel are good. However, I would like to see a stronger link to the literature throughout - there are many places where statements are made without obvious links to the supporting literature. My comments have been made chronologically - without line number it's difficult to indicate locations but I hope they are clear enough. IntroductionParagraph 1, I'm not quite sure how the influx of fish research is related to increased interest. Is this simply because of the frequent use of fish models in general?
final line, could do with reference to some examples. Table 1: Densing should be replaced with "increasing density of". Furthermore, schooling is a distinct behaviour and different to shoaling. I think this box requires a little more definition. Are there titles for the tables? While informative it's not immediately obvious what the point of the tables is. "This protocol is not performed in experiments with Zebrafish,"
Why not? Paragraph beginning "The main difference compared to other acute toxicity protocols is the objective of understanding the
mechanisms of action of test substances". I'm not sure I'm in entire agreement regarding this paragraph. While there are powerful tools to determine the behavioural effects of toxins, there's nothing suggested here which demonstrates that the method presented can also determine the mechanism of action itself. I think this needs to be more strongly demonstrated. First line of the section "3.3 The fish embryo test", reads: "The fish embryo test (FET) used to determine the toxicological toxicological action of a drug." Rewrite as, "... is used to determine..." and remove the second, "toxicological". The whole paragraph is a little grammatically messy, but this is a minor point, and the meaning is otherwise clear. However, to bring into line with the rest of Section 3, I think the authors should present some recommendations for when fish embryo tests are appropriate (compared to other tests). The authors suggest that behaviour is 10-1000 times more sensitive than lethality measures. Based on what? This needs a reference and a little bit of context I think. The paragraph for Section 4.1.2 Adults reads, "Most assays for locomotor behavior in larvae focus on changes in speed or distance. However, neurotoxic effects can frequently be observed as changes in posture or form of swimming movements [51,52]. Little and Finger demonstrated that the lowest toxicant concentration that produced changes in adult locomotor behavior lies between 0.1 and 5.0 percent of the lethal concentration". There is a bit of a logical jump here, in the first instance from the preceding paragraph about larvae which seemingly continues here. It's not clear when the authors are writing about adults specifically. Is it possible for changes in posture or form or swimming movements to be observed in larvae? If not, why not? Conclusion - I think this paragraph is, as presented, a little weak and actually quite easily missed. I think the conclusion/summary needs to be a little bit more thorough and cover the range of arguments presented in the review. I also feel that mention of the 3Rs is somewhat perfunctory. It's the first time it's mentioned in the entire review, so it makes me wonder why it's there. Are the authors attempting to place their suggestions within the context of 3Rs research, or (as seems to be the case) putting the phrase in there in an attempt to quickly and easily highlight the impact of the work laid out in the review. If the former, then there is no satisfactory build up to the philosophy of 3Rs research. Indeed, the 3Rs philosophy could be (and perhaps should be) a cornerstone of this review, providing the context by which we should be considering alternative appropriate methodologies to assess toxicological studies.
So I think this final paragraph needs a lot of work and I suggest that consideration of the underlying rationale of the review be considered in light of it. This shouldn't necessarily require an overhaul of the paper but, rather, a more cohesive demonstration of why it's important.
Author Response
The authors are thankful for the reviewer’s comments, which were insighftul and helped the manuscript improve considerably. The authors are also thankful for the reviewer takin his/her time to review our manuscript, especially in a time of pandemic. In what follows, we attempt to address the concerns raised by the reviewer. Changes in the text are marked in red.
Overall this is an interesting review of toxicological methods, and the authors provide a convincing argument for using alternative procedures than those currently popular. Most of my comments are relatively minor in nature and deal with structure of the review, rather than its contents which I feel are good. However, I would like to see a stronger link to the literature throughout - there are many places where statements are made without obvious links to the supporting literature.
My comments have been made chronologically - without line number it's difficult to indicate locations but I hope they are clear enough.
Introduction
Paragraph 1, I'm not quite sure how the influx of fish research is related to increased interest. Is this simply because of the frequent use of fish models in general?
The first and second paragraphs of the Introduction were expanded to include references to possible reasons why fish (especially zebrafish) gained prominence in the area of behavioral and neural toxicology. These include not only increased interest and, as the referee suggested, increased use of fish in research in general, but also increased availability of laboratory assays, pressures from special interests groups, and so forth. This is made explicit in lines 60-66:
“he field of ecotoxicological research using fish species gained much traction in the last 20 years [12]. While behavioral testing has been common for a longer period (at least since the 1970s; e.g., Rand’s 1985 review [13]), the increased availability of laboratory assays and protocols, as well as pressures from special interests groups and increasing awareness from regulatory agencies that lethal endpoints are not appropriate for ecotoxicology, have brought renewed interest in the field in a way that sublethal behavioral and physiological effects are now more common in the area than the usual protocols [14].”
Final line, could do with reference to some examples.
References and examples were added at the end of the first paragraph, as follows:
Examples of contaminants that have been shown to change cautious exploratory behavior include atrazine [4], methylmercury [5,6], PCB126 [7], dimethyl sulfoxide [8], and copper [9]. Ammonia [10] and IPBC (3-iodo-2-propynyl-N-butyl carbamate)[11] have been shown to impact antipredatory and alarm responses.
Table 1: Densing should be replaced with "increasing density of". Furthermore, schooling is a distinct behaviour and different to shoaling. I think this box requires a little more definition.
Are there titles for the tables? While informative it's not immediately obvious what the point of the tables is.
-These terms were compiled from OECD guideline 203. The reference to shoaling and schooling appears as “shoaling/schooling” in that guideline. However, to increase clarity and explicitly differentiate shoaling and schooling, the box was changed to “Loss or increased density of schooling or shoaling behavior”
-Titles that were missing were added to all tables, explaining their meaning and usefulness as “fast references” in the text.
"This protocol is not performed in experiments with Zebrafish," Why not?
Zebrafish are too small to implant electrodes to follow changes in cardiovascular activity, and it is impossible to draw blood in significant amounts to analyze, e.g., electrolytes. This has been explained in the session, as follows:
Lines 154-156: To the best of our knowledge, this protocol has never been performed in experiments with zebrafish, due to size limitations on implantation of electrodes and blood testing, but is done in other species of fish
Lines 164-166: While size limitations on using zebrafish make it difficult to apply FATS to zebrafish and other small teleosts, cardiotoxicity has been assessed in vivo in D. rerio larvae (e.g., ref. [29]).
Paragraph beginning "The main difference compared to other acute toxicity protocols is the objective of understanding the mechanisms of action of test substances". I'm not sure I'm in entire agreement regarding this paragraph. While there are powerful tools to determine the behavioural effects of toxins, there's nothing suggested here which demonstrates that the method presented can also determine the mechanism of action itself. I think this needs to be more strongly demonstrated.
The term “mechanism of action” is used in the FATS papers to refer to general pharmacological classes based on effects on cardiorespiratory phenomena in humans. The term is not entirely correct from a pharmacological point of view (i.e., which are the targets of these drugs). To address this issue, the term has been changed to “mode of action” throughout Section 2.2. Thus, the second paragraph of this section (lines 167-179) now reads:
“The main difference compared to OECD acute toxicity protocols is that FATS’ purported objective is that of understanding the pathophysiological mode of action of test substances. The understanding of the mode of action is not only thought to represent particular events at the molecular level, but also to understand the causal mechanisms in specific toxins; for this there are direct and indirect measures of the biological response of the organism produced for each substance. For example, the combination of quantitative structure-activity relationship (QSAR) information with the effects of toxicants on FATS endpoints suggests “cardiorespiratory syndromes” which are associated with AChE inhibitors, convulsants, narcotics, respiratory blockers, respiratory membrane irritants, and uncouplers to be correctly predicted [30–32], a promising avenue that points to FATS as an interesting toxicological screen. These qualities suggest that FATS assays have good predictive validity - that is, the assays are capable of discriminating between different toxic effects based on mode of action.”
First line of the section "3.3 The fish embryo test", reads: "The fish embryo test (FET) used to determine the toxicological toxicological action of a drug." Rewrite as, "... is used to determine..." and remove the second, "toxicological". The whole paragraph is a little grammatically messy, but this is a minor point, and the meaning is otherwise clear. However, to bring into line with the rest of Section 3, I think the authors should present some recommendations for when fish embryo tests are appropriate (compared to other tests).
The Section on FET has been completely rewritten, expanding it to include uses and issues. The Section now reads (lines 182-221):
The fish embryo toxicity test (FET) is widely used to determine the toxicological and teratological action of a drug or toxicant [33,34]. Doses are applied at different stages of the zebrafish (Danio rerio) embryo. The fish are fertilized, the eggs collected and exposed to the chemical test for a period of 96 hours. After the application, observations are made on the effect of the lethality of the drug on the embryos. FET has been optimized, standardized, and validated during an OECD validation study and adopted as OECD 236 as a test to assess embryotoxicity and teratogenicy [34]. The test has a good correlation with results from acute toxicity tests (OECD 203), suggesting at least an alternative from the refinement point of view if used to find ranges of toxic concentrations [33]. Some limitations of the test have been pointed [33]: that it is unsure whether biotransformation of toxic compounds take place in embryos at such early stage; and that the protection afforded by the chorion would not allow highly lipophilic and/or high molecular weight substances to access the embryo. Moreover, at its simplest version, FET does not include sublethal endpoints, which severely limit its use as a toxicological screen. Importantly, there is some evidence that compounds which show neurotoxicity in adults are weakly neurotoxic in the FET [35]; however, adding analysis of locomotor activity (see 3.1.1, below) has been shown to improve the ability of FET to detect neurotoxic compounds [35].
From an ecotoxicological point of view, two limitations of FET arise. Stelzer et al. [36] showed that FET is less sensitive than US Environmental Protection Agency method 2000.0 [37], which uses larvae instead of embryos, to detect toxic effects of untreated hospital effluents. Again, adding sublethal endpoints (immobility, nonhatching, and pericardial edema) increased the sensitivity of FET to detect toxic effects of water effluents [36]. A second limitation is that effects of toxicants, such as endocrine disruptors or potentially neurotoxic agents, might not be seen immediately, but, because these compounds affect developmental molecules, are observed after a prolonged interval (i.e., developmentally delayed effects). In that sense, exposure to toxicants at the egg or embryo stages might not induce gross changes in mortality of anatomy in embryos, but can produce important changes in adult neurobehavioral endpoints much later in life. Examples of substances which have been shown to produce delayed neurobehavioral toxic effects include PCB126 [7] and chlorpyrifos [38,39].
As with the acute toxicity test, the utility of FET in ecotoxicological research could be increased by adding it in more complex designs. For example, animals could be exposed after fertilization, and the usual endpoints of FET (mortality, gross anatomical changes), as well as sublethal endpoints (immobility, nonhatching, and pericardial edema) observed up to 96 h after exposure (i.e., up to 4 dpf). At later stages (5-7 dpf), behavioral observations could be made (see 3, below), and ecologically-relevant behaviors could also be recorded at adult age. Thus, not only screens could be refined and animal use reduced, but the effects of potentially neurotoxic substances would be observed across life stages.
The authors suggest that behaviour is 10-1000 times more sensitive than lethality measures. Based on what? This needs a reference and a little bit of context I think.
The context for that is estimates made by the Peterson et al. and Gerhardt papers (refs. 40 and 45). These references have been added to the paragraph.
The paragraph for Section 4.1.2 Adults reads, "Most assays for locomotor behavior in larvae focus on changes in speed or distance. However, neurotoxic effects can frequently be observed as changes in posture or form of swimming movements [51,52]. Little and Finger demonstrated that the lowest toxicant concentration that produced changes in adult locomotor behavior lies between 0.1 and 5.0 percent of the lethal concentration". There is a bit of a logical jump here, in the first instance from the preceding paragraph about larvae which seemingly continues here. It's not clear when the authors are writing about adults specifically. Is it possible for changes in posture or form or swimming movements to be observed in larvae? If not, why not?
The phrasing has been changed to reflect the fact that both larvae and adults show changes in posture and form that can be observed and quantified, but adult behavior is expected to be more complex and easier to observe. This now reads as follows:
Most assays for locomotor behavior in larvae focus on changes in speed or distance [60]. However, neurotoxic effects can frequently be observed as changes in posture or form of swimming movements in both larvae and adults [78,79]. More fine-grained assays for larvae were described above (see Table 3). However, adult behavior is putatively more complex than larval behavior, and changes in movement patterns are more readily observable due to larger size.
Conclusion - I think this paragraph is, as presented, a little weak and actually quite easily missed. I think the conclusion/summary needs to be a little bit more thorough and cover the range of arguments presented in the review. I also feel that mention of the 3Rs is somewhat perfunctory. It's the first time it's mentioned in the entire review, so it makes me wonder why it's there. Are the authors attempting to place their suggestions within the context of 3Rs research, or (as seems to be the case) putting the phrase in there in an attempt to quickly and easily highlight the impact of the work laid out in the review. If the former, then there is no satisfactory build up to the philosophy of 3Rs research. Indeed, the 3Rs philosophy could be (and perhaps should be) a cornerstone of this review, providing the context by which we should be considering alternative appropriate methodologies to assess toxicological studies.
So I think this final paragraph needs a lot of work and I suggest that consideration of the underlying rationale of the review be considered in light of it. This shouldn't necessarily require an overhaul of the paper but, rather, a more cohesive demonstration of why it's important.
The Conclusion paragraph was overhauled, as the focus of the paper is more narrowly defined:
In summary, the power of behavioral experiments in the laboratory can be tapped to expand the reach of ecotoxicological approaches by increasing the number of tests with a domain-based mindset. This does not mean that lethal (acute) tests should be abandoned, given their importance in regulatory agencies and policies regarding, e.g., pesticides. However, using behavioral tests (including the zebrafish bioassays briefly described in this review) can add to the options of sublethal tests that are not only more sensitive than lethal tests, but more likely to detect biologically relevant effects that can affect the mesocosm (population, species, community, ecosystem). The long history of research on sublethal toxicity in fish, and the uneasy relationship between the fields that compose behavioral ecotoxicology (ethology, behavioral neuroscience/pharmacology, and toxicology), need to be put in an integrative framework. To do that, the problem of specificity and the problem of translatability (that is, whether increased sensitivity and specificity translate to better predictions of effects at the population and higher levels, and whether these predictions impact environmental policies) need to be addressed both at the ethical and empirical frames. The hope of using the full potential of the zebrafish can only be approached if these steps are taken first.

Reviewer 2 Report
In this study, it was reviewed (sub-lethal) effects of toxicants on neurobehavioral traits of different life cycle stages of zebrafish, to demonstrate that these features are highly sensitive to low nominal concentrations. Authors aim to call attention to the relevance of expanding (eco)toxicological data associated to these features, beyond lethality (mortality and teratogenicity) and/or crude morphological endpoints. It is recognized the interest and relevance of the subject. Although relatively satisfactory written, some parts were identified of being confusing in their message. Reading is not always easy to follow. As this review focus specifically on different life cycle of zebrafish, the title should reflect it and avoid using a general reference as "Fish models (...)". References are lacking as support of statements, and references list is composed of quite old documents. Revision by a native English speaker is recommended. Figures and Tables require adjustments. Detailed questions/concerns were highlighted through the manuscript document (please see comments attached).

Author Response
The authors are thankful for the reviewer’s comments, which were insighftul and helped the
manuscript improve considerably. The authors are also thankful for the reviewer taking his/her time
to review our manuscript, especially in a time of pandemic. In what follows, we attempt to address
the concerns raised by the reviewer. Changes in the text are marked in red.
I am not sure who this review would be useful for; it is far too superficial to make a meaningful contribution. I think the review would be better focused on just reviewing recent applications of toxicity testing – there are upwards of 75 papers a year currently published on the issue and reviewing current approaches, applications and limitations would be very helpful. The paper would benefit from more focus, more detail, and more comparisons with traditional endpoints to give a perspective on relative sensitivity.
The review is very superficial, and starts with a very general introduction on model organisms, a very general introduction to basic acute toxicity testing, and a very superficial overview of one set of McKim’s experiments in the 1980s. The fish embryo test (p 6) is again a very superficial view of one application of the test.
The introduction and the section on acute toxicity testing were meant to present to readers the idea that using fish in toxicity testing is not a new thing, and even the proposal of using behavior dates from the 1970s. However, the increased interest in zebrafish behavioral assays provide an opportunity to make more interesting approaches between the fields of behavioral neuroscience and toxicology. The sections on FATS and FET have been considerably expanded to provide more context and applications, as well as to suggest acute toxicity tests can be used in a wider framework that also considers larval and adult behavioral toxicity. These changes have been marked in red in the text, and are too extensive to summarize in this response. As an example, however, take the changes that have been made to the session on FET (lines 182-221):
The fish embryo toxicity test (FET) is widely used to determine the toxicological and teratological action of a drug or toxicant [33,34]. Doses are applied at different stages of the zebrafish (Danio rerio) embryo. The fish are fertilized, the eggs collected and exposed to the chemical test for a period of 96 hours. After the application, observations are made on the effect of the lethality of the drug on the embryos. FET has been optimized, standardized, and validated during an OECD validation study and adopted as OECD 236 as a test to assess embryotoxicity and teratogenicy [34]. The test has a good correlation with results from acute toxicity tests (OECD 203), suggesting at least an alternative from the refinement point of view if used to find ranges of toxic concentrations [33]. Some limitations of the test have been pointed [33]: that it is unsure whether biotransformation of toxic compounds take place in embryos at such early stage; and that the protection afforded by the chorion would not allow highly lipophilic and/or high molecular weight substances to access the embryo. Moreover, at its simplest version, FET does not include sublethal endpoints, which severely limit its use as a toxicological screen. Importantly, there is some evidence that compounds which show neurotoxicity in adults are weakly neurotoxic in the FET [35]; however, adding analysis of locomotor activity (see 3.1.1, below) has been shown to improve the ability of FET to detect neurotoxic compounds [35].
From an ecotoxicological point of view, two limitations of FET arise. Stelzer et al. [36] showed that FET is less sensitive than US Environmental Protection Agency method 2000.0 [37], which uses larvae instead of embryos, to detect toxic effects of untreated hospital effluents. Again, adding sublethal endpoints (immobility, nonhatching, and pericardial edema) increased the sensitivity of FET to detect toxic effects of water effluents [36]. A second limitation is that effects of toxicants, such as endocrine disruptors or potentially neurotoxic agents, might not be seen immediately, but, because these compounds affect developmental molecules, are observed after a prolonged interval (i.e., developmentally delayed effects). In that sense, exposure to toxicants at the egg or embryo stages might not induce gross changes in mortality of anatomy in embryos, but can produce important changes in adult neurobehavioral endpoints much later in life. Examples of substances which have been shown to produce delayed neurobehavioral toxic effects include PCB126 [7] and chlorpyrifos [38,39].
As with the acute toxicity test, the utility of FET in ecotoxicological research could be increased by adding it in more complex designs. For example, animals could be exposed after fertilization, and the usual endpoints of FET (mortality, gross anatomical changes), as well as sublethal endpoints (immobility, nonhatching, and pericardial edema) observed up to 96 h after exposure (i.e., up to 4 dpf). At later stages (5-7 dpf), behavioral observations could be made (see 3, below), and ecologically-relevant behaviors could also be recorded at adult age. Thus, not only screens could be refined and animal use reduced, but the effects of potentially neurotoxic substances would be observed across life stages.
P 2 para 2 Fish research has had traction for more than 50 years – trying to emphasize that this is rapid – behavioural testing has been common since the 1970s but has received renewed attention. The publishing rate has tripled over the last decade but it is not new or revolutionary and I would not characterize the movement as “rapidly moving beyond mortality and teratogenicity”
This section has been rewritten to reflect the fact that using behavior is not new in the field, but that the search for sublethal endpoints (behavior included) received renewed attention due to many factors, including changes in policy and the emergence of zebrafish as a model organism in behavioral neuroscience. The session now reads (lines 60-66):
The field of ecotoxicological research using fish species gained much traction in the last 20 years [12]. While behavioral testing has been common for a longer period (at least since the 1970s; e.g., Rand’s 1985 review [13]), the increased availability of laboratory assays and protocols, as well as pressures from special interests groups and increasing awareness from regulatory agencies that lethal endpoints are not appropriate for ecotoxicology, have brought renewed interest in the field in a way that sublethal behavioral and physiological effects are now more common in the area than the usual protocols [14].
P 6 last paragraph – it is easy to say that these relationships are philosophically possible, but reviewing evidence that has demonstrated that behavioural assays have been predictive would be much more useful
P 7 – in terms of advantages a) a review of how and when it is predictive would be useful, b) the review would benefit from considering the implications of increasing sensitivity >1000-fold in terms or regulatory perspectives, and c) considering when early warning is useful would also be appropriate.
The review of behavioural responses would benefit from a comparison with known guidelines, and traditional toxicity responses to give a picture of the relative sensitivity. The standard for interpreting impact has been an impact on growth, survival and reproduction; need to demonstrate that correlation to higher endpoints or could be over-protective.
A whole new session (session 5, lines 434-484) was added to address some of these issues, including the question of whether behavioral assays are predictive of effects at higher levels, as well as the impacts of increasing sensitivity have in regulatory and policy work. The authors understand that the creation of policies is the final aim of ecotoxicology as a field, but the review was not intended to discuss these issues. Nonetheless, the authors agree with the referee that these issues are urgent, and call attention to the need for researchers (especially zebrafish researchers) to also focus on that.
P 13 section 5 first sentence – I think it means Section 4.
The reference to section 5 was corrected to Section 4, as pointed by the reviewer.
I am not sure how the behavioural testing complies with reducing animal use – what happens to the animals after the experiment?
Many guidelines for the use of fish in experimental research (including Brazilian guidelines) suggest that animals are donated after results, thus reducing death. Nonetheless, as references to the 3Rs were limited to the conclusion, and – in part due to the important contributions made by the referee – the direction of the paper is now more narrowly defined, the conclusion has changed, and references to the 3Rs were removed.
Reference 1 and 38 are missing a year pf publication
References have been corrected now.

Reviewer 3 Report
I am not sure who this review would be useful for; it is far too superficial to make a meaningful contribution. I think the review would be better focused on just reviewing recent applications of toxicity testing – there are upwards of 75 papers a year currently published on the issue and reviewing current approaches, applications and limitations would be very helpful. The paper would benefit from more focus, more detail, and more comparisons with traditional endpoints to give a perspective on relative sensitivity.
- The review is very superficial, and starts with a very general introduction on model organisms, a very general introduction to basic acute toxicity testing, and a very superficial overview of one set of McKim’s experiments in the 1980s. The fish embryo test (p 6) is again a very superficial view of one application of the test.
- P 2 para 2 Fish research has had traction for more than 50 years – trying to emphasize that this is rapid – behavioural testing has been common since the 1970s but has received renewed attention. The publishing rate has tripled over the last decade but it is not new or revolutionary and I would not characterize the movement as “rapidly moving beyond mortality and teratogenicity”
- P 6 last paragraph – it is easy to say that these relationships are philosophically possible, but reviewing evidence that has demonstrated that behavioural assays have been predictive would be much more useful
- P 7 – in terms of advantages a) a review of how and when it is predictive would be useful, b) the review would benefit from considering the implications of increasing sensitivity >1000-fold in terms or regulatory perspectives, and c) considering when early warning is useful would also be appropriate.
- The review of behavioural responses would benefit from a comparison with known guidelines, and traditional toxicity responses to give a picture of the relative sensitivity. The standard for interpreting impact has been an impact on growth, survival and reproduction; need to demonstrate that correlation to higher endpoints or could be over-protective.
- P 13 section 5 first sentence – I think it means Section 4.
- I am not sure how the behavioural testing complies with reducing animal use – what happens to the animals after the experiment?
- Reference 1 and 38 are missing a year pf publication
Author Response

(The authors gave the same response as above.)

Reviewer 4 Report
See attached file

Author Response
The authors are thankful for the reviewer’s comments, which were insighftul and helped the
manuscript improve considerably. The authors are also thankful for the reviewer taking his/her time
to review our manuscript, especially in a time of pandemic. In what follows, we attempt to address
the concerns raised by the reviewer. Changes in the text are marked in red.
The authors have placed their review in the context of acute toxicity testing, as appears from the
title, the Abstract and the Introduction. This is not correct. Acute toxicity tests are used for testing
the toxicity of chemicals for humans and animals. The protocols are described in for instance the
OECD guidelines (these are cited by the authors) and the tests have teratogenicity or mortality as
endpoints, and the results are expressed in LC50 or LC96 concentrations etc, indicating the
concentrations that kill 50 or more percent of the animals in a given rather short period of time. Such tests have a legal basis and are obligatory for all new chemicals including pesticides, and the results of these tests determine whether and how the chemicals can be put into the market. Such acute toxicity tests can never be replaced by the neural and behavioural tests for sublethal toxicity
described in this review.
The sublethal toxicity tests are used for testing the effects of low to very low levels of toxicants in for instance the food for human consumption or for consumption by animals in agriculture or
aquaculture (toxicology and health). A second field of application for sublethal tests is the effect on
the environment, including the testing of water quality in natural waters or in aquaculture
(ecotoxicology). The neural and behavioural tests belong to this field of sublethal toxicity testing.
The manuscript was not intended to suggest that lethal endpoints (acute toxicity tests) are inadequate and should be replaced by neurobehavioral assays. The referee correctly point out that the uses of acute toxicity testing and sublethal endpoints are different; indeed, throughout the manuscript we refer to the fact that sublethal endpoints are more sensitive, which only makes sense if one is looking to test effects of very low levels of toxicants. The authors fully agree with the referee in that this focus needs to be sharpened throughout the review. For example, in lines 224-235, we refer to behavioral tests as a “partition” of sublethal tests, and acknowledge the important literature on other parameters.
Thus:
-The title is not correct: the subtitle “Expanding beyond mortality and teratogenicity” refers to acute
toxicity. The neural and behavioural tests reported here are expanding the field of sublethal
toxicological tests. The title should be corrected.
The title has been changed to “Zebrafish models in neural and behavioral toxicology across the life stage”
- The title, Abstract and Introduction ignore the extensive litterature on existing sublethal tests on
fish. These tests concern physiological parameters in blood and other body fluids (ions, hormones
including stress hormones, metabolites including glucose levels etc.), histological parameters, and
biochemical parameters (effects on enzyme activity in blood, liver, intestine, brain..) etc. Although
sublethal toxicity is mentioned briefly, this litterature consisting of thousands of toxicology papers on fish only, and that have been reviewed many times, is missing completely. This omission should be repaired and the Abstract and Introduction as well as the final conclusion should be put in the
perspective of sublethal toxicology.
The authors agree that neurobehavioral testing is a subfield of sublethal toxicology. This has been acknowledged in Session 3 of the paper. However, we would also like to point out that this is not intended as a review of sublethal toxicity, but of fish behavioral models, focusing on zebrafish models. Thus, discussing more extensively the thousands of papers that the referee points to has already been made in reviews that have this specific aim, and falls beyond the scope of this manuscript.
-The advantages of neural and behavioural methods in sublethal toxicology should be explained by
comparing the results of these methods with the other methods for sublethal testing.
The advantages of using neural and behavioral methods have not been fully characterized since they were first proposed in the 1970s. However, by framing these behavioral bioassays in the larger context of ecotoxicology, linking them to predict effects at higher levels, we suggest that this could be an advantageous avenue, being careful to state that this has not been fully realized yet, and that the hypothesis that performance in a laboratory test could predict impacts on fitness is yet to be fully tested. Again, we would like to point out that the aim of the review is not to compare behavioral tests with physiological, biochemical, and histopathological endpoints; in fact, much of the current research on toxicology using zebrafish combines some of these parameters to attempt to integrate impacts on different organ systems, which ultimately reduces fitness.
Minor comment:
Legends to all the tables are missing. This makes the information of the Tables difficult to
understand.
Legends to the tables have been included. The authors are sorry for the mistake.

Round 2
Reviewer 2 Report
Self-plagiarism was detected regarding the current manuscript titled "Fish models in neural and behavioral toxicology: Expanding beyond mortality and teratogenicity" (title at first original version) . Please see at the link below:
https://europepmc.org/article/ppr/ppr93164
Author Response
This is certainly a mistake, as the referenced is precisely the preprint that was submitted at preprints.org - as indicated in the manuscript, as well as in the cover letter.
Reviewer 3 Report
I like this version much better. It is better organized but the message still needs clarification. I think it would benefit from narrowing the focus further still to a review on “Options for neural and behavioural toxicity tests using Zebrafish models”. Section 1 is largely unnecessary and Section 2 could be refocused to a general review of the re-emergence of behavioural bioassays and a review of recent reviews. Trying to force it into OECD protocols is not really relevant to the main message: behavioural bioassays have made a resurgence, and there have been a lot of applications using zebrafish creating a plethora of opportunities.
Line 46 – not appropriate use of “mesocosm” term
Line 63 – “regulatory agencies that lethal endpoints are not appropriate for ecotoxicology” is not something I have ever heard of – it would need some sort of supportive evidence. Toxicity testing is still a regulatory requirement in most jurisdictions. There may be requirements for supplemental testing but if lethal tests have been removed from regulatory approaches it would be worth noting. Or is the reference to their use for chemical screening, product registration, or some other use of toxicity testing. As far as I am aware, it is still widely used as a standard test for effluent testing.
Line 81-83 – incomplete sentence
Lines 227-228 – “However, as we argued above, there are many other toxic effects that are relevant for ecotoxicology” – I think these statements throughout need to get much more specific – ecotoxicology is many things – be specific as to what you are referring to – relevant for registering chemicals, screening chemicals, etc
Line 235 – figure caption is missing
Line 246-252 – another difficulty is interpreting the relevance to real world exposures and consequences.
Line 250 – starts to narrow down “ecotoxicology” to “mechanistic ecotoxicology” – is that the focus?
Line 264 – “clarify “as that”
Author Response
The authors are thankful for the reviewers comments. A major overhaul of the paper was attempted, removing Section 2 (Acute toxicity) to make the message more straightforward. Below, we answer the reviewers' concerns:
I like this version much better. It is better organized but the message still needs clarification. I think it would benefit from narrowing the focus further still to a review on “Options for neural and behavioural toxicity tests using Zebrafish models”. Section 1 is largely unnecessary and Section 2 could be refocused to a general review of the re-emergence of behavioural bioassays and a review of recent reviews. Trying to force it into OECD protocols is not really relevant to the main message: behavioural bioassays have made a resurgence, and there have been a lot of applications using zebrafish creating a plethora of opportunities.
Section 2 has been completely removed to narrow down the focus of the paper. We beg to disagree, however, on Section 1 being unnecessary: many researchers which work on behavioral toxicology (especially the newer generation that works with zebrafish) are not familiar with the modeling literature, and sometimes mistake both uses of fish as models. Moreover, since this is intended for a special issue on fish models in toxicology, we understand that bringing this literature to readers could be useful. Finally, since this section has only 4 paragraphs, we opted to maintain it in this version of the manuscript.
Line 46 – not appropriate use of “mesocosm” term
The term has been eliminated.
Line 63 – “regulatory agencies that lethal endpoints are not appropriate for ecotoxicology” is not something I have ever heard of – it would need some sort of supportive evidence. Toxicity testing is still a regulatory requirement in most jurisdictions. There may be requirements for supplemental testing but if lethal tests have been removed from regulatory approaches it would be worth noting. Or is the reference to their use for chemical screening, product registration, or some other use of toxicity testing. As far as I am aware, it is still widely used as a standard test for effluent testing.
The intended meaning was not lethal endpoins are not useful, but that they are not enough. We fully agree that, as written, the sentence does not convey it. The sentence has been changed to "increasing awareness from regulatory agencies that lethal endpoints are not sufficient for ecotoxicology"
Line 81-83 – incomplete sentence
The sentence has been changed to "The effects of a contaminant can be studied using fish behavior as a toxicological indicator"
Lines 227-228 – “However, as we argued above, there are many other toxic effects that are relevant for ecotoxicology” – I think these statements throughout need to get much more specific – ecotoxicology is many things – be specific as to what you are referring to – relevant for registering chemicals, screening chemicals, etc
The reference to "above" no longer makes sense, as Section 2 has been removed. Since the focus of the review is on screening chemicals and on mechanistic ecotoxicology, the sentence has been changed to "However, there are many other toxic effects that are relevant for ecotoxicological research, including its consequences on protocols for screening chemicals"
Line 235 – figure caption is missing
A caption has now been added.
Line 246-252 – another difficulty is interpreting the relevance to real world exposures and consequences.
This is further discussed in Section 4 (prev. Sec. 5). We added a paragraph referring this: "A corollary of predictive validity, of course, is that the model should be able to predict real world exposures and consequences. Most laboratory tests do not use real-world conditions; laboratory experiments isolate toxicants instead of using mixtures, often fail to use concentrations that are relevant in the wild, and focus on behaviors which are easy to manipulate or elicit. This is further discussed in Section 5."
Line 250 – starts to narrow down “ecotoxicology” to “mechanistic ecotoxicology” – is that the focus?
The focus of the review is chemical screening, but mechanistic ecotoxicology is also highly impacted by using neurobehavioral assays; the issue of specificity is very important, in that case.
Line 264 – “clarify “as that”
The term was changed to "they"
Reviewer 4 Report
The ms has been revised in a thoughtful fashion and has been significantly improved. The ms is more balanced: the toxicological tests with behavioural sublethal endpoints do no longer seem to be advocated as a replacement for the tests with lethal or teratogenic endpoints, but as a welcome and important extension of the toxicological toolbox. Still a few comments:
Major: Of the non-specific effects of toxicants, the effects of the general non-specific stress response are among the most important ones. These are not mentioned in the paper. These deserve at least a few lines in the paragraph starting with line 440, and/or with line 466. In fish any toxicant that affects the gill physiology (not only effects leading to hypoxia, but also effects disturbing the ion transport processes across the gills, which usually happens at much lower toxicant levels) immediately evoke a non-specific stress response, with typical behavioral aspects. This cannot be considered a "false positive"!.
Minor: Title: stages (and not stage)
Lines 81-83: A verb is missing. Perhaps this line should start: The effects of these contaminants can be studied using...
Author Response
Authors are thankful for the observations made by the reviewer. In special, a paragraph on non-specific stress responses, using alterations of gill physiology as an example, has been added (lines 374-387).